# Luminescence in Anion-Deficient Hafnia Nanotubes

**DOI:** 10.3390/nano13243109

**Published:** 2023-12-10

**Authors:** Artem O. Shilov, Robert V. Kamalov, Maxim S. Karabanalov, Andrey V. Chukin, Alexander S. Vokhmintsev, Georgy B. Mikhalevsky, Dmitry A. Zamyatin, Ahmed M. A. Henaish, Ilya A. Weinstein

**Affiliations:** 1NANOTECH Centre, Ural Federal University, 19 Mira St., 620002 Yekaterinburg, Russia; ao.shilov@urfu.ru (A.O.S.); rvkamalov@gmail.com (R.V.K.); m.s.karabanalov@urfu.ru (M.S.K.); a.v.chukin@urfu.ru (A.V.C.); a.s.vokhmintsev@urfu.ru (A.S.V.); zamyatin@igg.uran.ru (D.A.Z.); ahmed.henaish@science.tanta.edu.eg (A.M.A.H.); 2Institute of Geology and Geochemistry, Ural Branch of the RAS, Vonsovskogo Street, 15, 620110 Yekaterinburg, Russia; gosha622@gmail.com; 3Physics Department, Faculty of Science, Tanta University, Tanta 31527, Egypt; 4Institute of Metallurgy, Ural Branch of the RAS, Amundsena Street, 101, 620108 Yekaterinburg, Russia

**Keywords:** hafnium dioxide, nanotubes, oxygen deficiency, non-stoichiometric HfO_2_, direct bandgap, modified Tauc approach, low-temperature photoluminescence, PL decay

## Abstract

Hafnia-based nanostructures and other high-k dielectrics are promising wide-gap materials for developing new opto- and nanoelectronic devices. They possess a unique combination of physical and chemical properties, such as insensitivity to electrical and optical degradation, radiation damage stability, a high specific surface area, and an increased concentration of the appropriate active electron-hole centers. The present paper aims to investigate the structural, optical, and luminescent properties of anodized non-stoichiometric HfO_2_ nanotubes. As-grown amorphous hafnia nanotubes and nanotubes annealed at 700 °C with a monoclinic crystal lattice served as samples. It has been shown that the bandgap *E_g_* for direct allowed transitions amounts to 5.65 ± 0.05 eV for amorphous and 5.51 ± 0.05 eV for monoclinic nanotubes. For the first time, we have studied the features of intrinsic cathodoluminescence and photoluminescence in the obtained nanotubular HfO_2_ structures with an atomic deficiency in the anion sublattice at temperatures of 10 and 300 K. A broad emission band with a maximum of 2.3–2.4 eV has been revealed. We have also conducted an analysis of the kinetic dependencies of the observed photoluminescence for synthesized HfO_2_ samples in the millisecond range at room temperature. It showed that there are several types of optically active capture and emission centers based on vacancy states in the O_3f_ and O_4f_ positions with different coordination numbers and a varied number of localized charge carriers (V^0^, V^−^, and V^2−^). The uncovered regularities can be used to optimize the functional characteristics of developed-surface luminescent media based on nanotubular and nanoporous modifications of hafnia.

## 1. Introduction

At present, hafnium dioxide is highly sought after for creating new hardware components in nanoelectronics. Along with the usage of HfO_2_ as a gate dielectric in CMOS transistors [1], it is beneficial as a functional medium in designing memristor-cell-based memory devices to operate, relying on mechanisms involving induced defects of their intrinsic nature [2,3,4,5]. It is known that hafnia includes oxygen vacancies of various configurations and charge states as active electron-optical centers [6]. It is these vacancies that localize charges; thus, HfO_2_ is able to exhibit its own luminescence [7,8] and typical memristive behavior [2,3,9,10]. In addition, due to the high atomic mass of its cation, hafnia is an excellent solid-state matrix for doping with rare earth ions. This makes it possible to project up-to-date scintillation media and efficient light-emitting devices [8,9,11].

One of the methods for the targeted synthesis of hafnia with variable anionic non-stoichiometry is to develop low-dimensional structures with a high concentration of surface defects as a consequence of morphological features and non-equilibrium growth conditions. For producing HfO_2_ nanotubes (NTs), various physico-chemical methods can be utilized, such as atomic layer deposition into a matrix of nanoporous aluminum oxide [12] and a combination of electro-spinning and ion sputtering [13,14], as well as anodic oxidation of metal foil [15,16]. Whatever technique is used, the grown nanotubes are initially amorphous. When treated at high temperatures, their atomic structure passes into tetragonal and more stable monoclinic crystalline phases [17]. In some cases, maintaining the amorphous structure during the required annealingis arduous. In particular, to remove precursor residues, when synthesizing nanotubes by magnetron sputtering of a hafnium target onto organic PVP fibers, heating to 500 °C is required. In this case, HfO_2_ passes into a monoclinic phase [14].

Compared to other synthesis methods, electrochemical oxidation is a relatively simple method for growing nanotube arrays on a metal substrate. One of the first successful attempts to obtain anodized HfO_2_ nanotubes is described in [15], where a mixture of sulfuric acid and sodium fluoride is applied as an electrolyte. Manipulating the morphology of the synthesized oxide layer—solid → porous → nanotubular—can be performed by varying both the electrolyte composition and parameters of electrochemical synthesis, such as a voltage between the electrodes, current flowing through the solution, oxidation time, and temperature [16]. In turn, the geometric parameters and phase composition of NTs have a dramatic influence on the features of their electronic subsystem and various structure-sensitive capabilities [18,19,20].

To complete the description of the role of anionic vacancy centers in the processes of charge redistribution and transfer, it is necessary to dive deeper into the patterns of forming the energy structure of a material with intrinsic defects and complexes based on them. Conducting experimental research on the optical and luminescent properties of HfO_2_ NTs is extremely important for establishing the aspects of their energy spectrum caused by the morphology of the material. Currently, independent investigations of the optical and luminescent properties of bulk and thin-film hafnia samples doped with various ions, mainly with an amorphous and monoclinic structure, have been undertaken [8,19,21,22,23,24,25]. Simultaneously, examining the developed surface structures of hafnia could provide insight into the role of surface optically active centers and defects of various natures in the radiative and non-radiative relaxation processes of electronic excitations in HfO_2_ nanotubes. The goal of the present work is to analyze the characteristics of the luminescent response of nanotubular structures of hafnia with an atomic deficiency in the anion sublattice.

## 2. Materials and Methods

### 2.1. Samples

In this work, we study the properties of three samples: nanotubular hafnia (as-grown and annealed NTs) and hafnia nanopowder. The nanotubular samples were synthesized using electrochemical anodization of hafnium foil (HFI-1 grade, 99.9% purity, MetallKomplekt LLC, Moscow, Russia), where Hf acted as the anode and stainless steel was used as the cathode. The anodization was carried out in a solution containing NH_4_F 0.5 wt% (98.5% purity, Klassik LLC, Moscow, Russia), distilled H_2_O 2 wt%, and ethylene glycol (99.5% purity, EKOS-1 JSC, Moscow, Russia). The synthesis was performed under a constant voltage of 40 V for 4 h. The synthesized samples contained carbon and fluorine as a part of the electrolyte used. It is known that the precursor residues can be removed from the anodized nanotubes of transition metals after heating to 300 °C [26]. Such treatment was executed in our work to obtain the as-grown sample. Additionally, the NTs were annealed in air at 700 °C for 2 h per annealed sample. The temperature treatments were performed at a heating rate of 10 °C/min and following natural cooling. Crystalline hafnia nanopowder (HFO-1 grade, TU 48-4-201-72) was used as a reference structure [7].

### 2.2. Experimental Techniques

The morphology was investigated using a JEOL JEM-2100 (Tokyo, Japan) transmission electron microscope (TEM) and a Carl Zeiss SIGMA VP (Oberkochen, Germany) scanning electron microscope (SEM) with an Oxford Instruments X-Max 80 (Abingdon, UK) module for energy dispersive analysis (EDS) to determine the chemical composition of the sample. To obtain SEM images, the accelerating voltage was set to 3 kV. X-ray diffraction (XRD) was analyzed using a Shimadzu XRD-7000 diffractometer (Tokyo, Japan). XRD patterns were measured for the 2θ range from 10° to 85° with 0.06° step, the diffractometer operating at 40 kV, and 30 mA with CuKα radiation using a monochromator. The Raman spectrum was measured with a Horiba LabRAM HR800 Evolution spectrometer (Palaiseau, France) in the range of 50–900 cm^−1^. Raman spectra were obtained with He-Ne 632.8 nm excitation (10 mW at the sample surface) using a diffraction grating with 600 gr/mm in the optical pathway. FTIR characterization was performed using a Bruker Vertex 70 spectrometer (Ettlingen, Germany) with an integrating sphere attachment. Hafnia samples were mixed with KBr powder as a diluent in order to obtain diffuse reflectance spectra in the 400–4000 cm^−1^ range. FTIR spectra were analyzed using Kubelka–Munk formalism.

Diffuse reflectance spectra (DRS) were measured in the 210–850 nm range with a step of 0.1 nm, using a two-beam spectrophotometer Shimadzu UV-2450 (Tokyo, Japan) and integrating sphere ISR-2200 (Shimadzu, Tokyo, Japan) attachment. Barium sulfate powder was used as a white body reference.

Cathodoluminescence (CL) spectra in the 250–1000 nm range were obtained with a Jeol JSM 6390LV (Tokyo, Japan) scanning electron microscope (accelerating voltage was set to 20 kV) using a Horiba H-CLUE i550 spectrometer (Palaiseau, France) equipped with a CCD detector and grating with 150 gr/mm. Photoluminescence (PL) measurements were taken using an Andor Shamrock SR-303i-B spectrograph (Belfast, UK) with a NewtonEM DU970P-BV-602 CCD recording array (Andor Technologies, Belfast, UK). A DTL-389QT ultraviolet laser (Laser-Compact Group, Moscow, Russia) with a wavelength of 263 nm (4.71 eV) was used as a photoexcitation source. PL excitation spectra and kinetic dependencies were recorded using a Perkin Elmer LS 55 spectrometer equipped with a xenon source. During measurements of the PL decay kinetics, the delay after turning off the Xe lamp was 0.1 ms and was increased to 60 ms in steps of 0.1 ms. The signal recording time amounted to 12.5 ms with a total duration of one measurement cycle of 80 ms. In the recording channel, the monochromator slit width was 20 nm, whereas in the excitation channel it was 10 nm. The luminescence was excited by 243 nm (5.1 eV) photons for decay curve registration. To plot the obtained luminescence spectra against photon energy, a correction was performed, which is necessary in cases where diffraction grating with linear dispersion in wavelengths is used [27]. Measurements of CL and Raman spectra were carried out at the Common Use Center “Geoanalyst” (Institute of Geology and Geochemistry of the Ural Branch of RAS, Ekaterinburg).

## 3. Results and Discussion

### 3.1. Electronic Microscopy

Figure 1 shows SEM and TEM images of the grown hafnia nanotubular arrays. The length and average diameter of the synthesized nanotubes were found to be 10 ± 3 μm and 46 ± 7 nm, respectively. The diameter distribution depicted in histogram 1d obeys a lognormal law with parameters μ = 3.83 and σ = 0.16. A chemical analysis conducted via the EDS method detected no impurities from heavy elements. After annealing up to 300 and 700 °C, only hafnium and oxygen were observed in the EDS spectra. According to the obtained data, the O/Hf ratio varies in a range between 1.78 and 1.91 in different regions of the samples, thus confirming the deficiency in the anion sublattice.

### 3.2. Structural Phase Analysis

Figure 2a outlines X-ray diffraction (XRD) analysis data for nanotubular structures. For as-grown NTs, a halo is observed in the range of 20–38 °C, and therefore, it can be argued that these structures are amorphous. In the process, the most intense peaks match a hafnium foil utilized for growing the oxide layer. In addition, there are several peaks indicating small inclusions of cubic and monoclinic phases of HfO_2_. When annealed, the samples exhibit no halo, and hafnia is in the monoclinic phase. For comparison, Figure 2a shows the diffraction pattern of monoclinic HfO_2_ nanopowder.

Raman and IR spectroscopy confirm the structural transition to a monoclinic crystalline phase, observed for the nanotubes after annealing—the data are presented in Figure 2b and Figure 2c, respectively. The Raman spectrum revealed seventeen active vibrational A_g_ and B_g_ modes. In the present research, the IR spectrum of hafnia nanotubes was measured for the first time, as shown in Figure 2c. Table 1 lists the phonon energies of IR-active vibrational modes in comparison with known data for thin films [28,29,30] and nanoparticles synthesized via precipitation [31] and the sol–gel method [32]. An analysis of the measured IR spectrum using the predicted data in [33] made it possible to identify five modes; see Figure 2c and Table 1. Additionally, there are maxima experimentally held fixed both in the nanotubes and in the monoclinic hafnia nanopowder; however, group theoretical analysis [33] did not determine their type (see Table 1). In addition, we observe a weak absorption within 1450–1600 cm^−1^ (see Figure 2c) which could be connected with H-O-H angular bending modes [34]. For as-grown nanotubes with an amorphous structure, the Raman and IR spectra contained no active vibrational modes.
Figure 2Results of XRD analysis (**a**) and Raman (**b**) and IR (**c**) spectroscopy for synthesized HfO_2_ nanotubes in comparison with data for HfO_2_ nanopowder with a monoclinic structure. The XRD and Raman data of the hafnia nanopowder were measured in Shilov et al. [7]. “?”—IR maxima which are not assigned to specific vibrational modes in calculations [33].
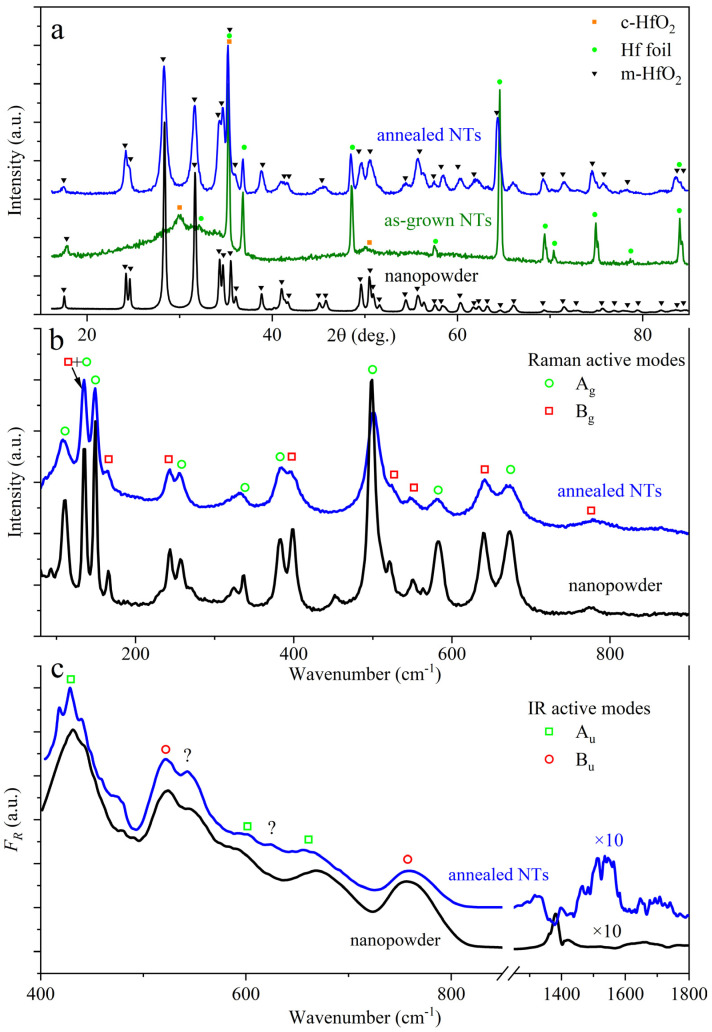


### 3.3. Estimation of the Bandgap Width

Using Kubelka–Munk formalism [7,35], we analyzed diffuse reflectance spectra recorded at room temperature (experimental data are depicted in Figure 3a). The resulting absorption spectrum *F_R_*(*h*ν) (see Figure 3b) includes a broad Gaussian-shaped shoulder that emerged because of oxygen vacancies: their absorption lies within energy region of *h*ν > 3 eV [36].

In order to correctly estimate the bandgap width using the Tauc approach, it was necessary to extract the optical absorption caused by active defects within a region of the spectral shoulder. The latter was approximated via a Gaussian curve (see the dashed curves in Figure 3b) and subtracted from the optical absorption spectra. Figure 3c includes revised curves *F_R_*(*h*ν) plotted in Tauc coordinates for direct allowed band-to-band transitions [37]. In the frame of the approach used, the optical gap width is *E_g_* = 5.65 ± 0.05 eV for as-grown and 5.51 ± 0.05 eV for annealed nanotubes. It is worth stressing that the above evaluations were pioneered for nanotubular HfO_2_ and that no independent data were found that would allow for comparison. A similar redshift of the bandgap was previously observed with hafnia thin films in [38]. The authors claim that the indirect bandgap in amorphous as-grown films is 5.60 eV, which decreases as the annealing temperature grows, reaching 5.13 eV for thin films annealed at 600 °C with a monoclinic crystal structure. Nevertheless, the values obtained are reconciled with previous estimates of direct *E_g_* for amorphous and monoclinic hafnia in various structural modifications, such as nanopowder (in range of 5.5–5.8 eV) [7,25,39,40], thin films (5.3–5.8 eV) [41,42,43], bulk single crystal (5.89 eV) [44], etc.

In hafnia, we have previously detected an intrinsic absorption edge formed via direct and indirect allowed band-to-band transitions [7,25]. However, due to the distortion of the intrinsic absorption edge in the low-energy region, we did not succeed in determining indirect optical gap width for as-grown and annealed NTs.

### 3.4. Spectral Characteristics of Luminescence

When photo- and cathodoluminescence in the samples under study were measured, a blue-green emission was observed at different temperatures (see Figure 4). This is characteristic of hafnia in various morphological modifications such as thin films [6,23,42,45], nanopowders [7,46,47], nanocrystals [48,49], and bulk single crystals [44]. It should be noted that cooling the nanotubes to a temperature of 10 K has almost no influence on the broad Gaussian-shaped bands on the PL spectra (see inset in Figure 4). It is known that the observed emission is associated with the presence of oxygen vacancies in hafnia [6,36,46,47]. Table 2 provides the values of the maximum energies (E_max_) and full widths at half maximum (FWHM) calculated during the approximation of the luminescence spectra via Gaussian curves, as well as the estimates of the intensity ratio at different temperatures.

The photoluminescence excitation spectra (PLE) of hafnia nanotubes can be seen in Figure 5. They contain four maxima, and their positions are almost unchanged after high-temperature treatment. The peaks at 4.6, 4.8 and 5.1 eV fall into the optical absorption region for oxygen vacancies [36] (see also the spectral shoulder in Figure 3b), and the 5.5 eV maximum rests in the region of optical band-to-band transitions near the intrinsic absorption edge (see Figure 3). The comparison of normalized PL spectra in Figure 5 points to the fact that the luminescence band for as-grown NTs at room temperature is slightly wider and insignificantly shifted to lower energies in comparison with a similar peak for annealed NTs.

Some papers [13,14] have investigated the photoluminescence of HfO_2_ nanotubes produced via magnetron sputtering of a hafnium target onto polyvinylpyrrolidone nanofibers doped with rare earth ions. According to [13], upon excitation via laser irradiation at 325 nm, emission with a maximum of 2.9 eV is observed. It is blue-shifted relative to our data obtained from excitation at 263 nm; see Table 2. The shift of the emission maximum may be associated with different PL excitation energies, as well as with different methods of synthesis of the nanostructures at hand. In particular, the average diameter of nanotubes in [13,14] is 200 nm, which is higher by a factor of 4 than the similar magnitude for the NTs grown via anodic oxidation in our work.

We were the first to measure the cathodoluminescence (CL) spectra of hafnia nanotubes, see Figure 4d and Table 2. It can be seen that the positions of the maxima in the PL and CL spectra virtually coincide. In addition, when subjected to high-temperature annealing, the samples exhibit a narrowing and a slight shift of the emission bands towards the short-wavelength region of the spectrum. Moreover, in contrast to the PL response, the CL spectrum for amorphous nanotubes has a more complicated shape that cannot be approximated by a single Gaussian. Since there are no independent experimental data for nanotubes, we can compare only our results with the CL spectra for monoclinic nanopowder and thin films of non-stoichiometric-in-oxygen sublattice hafnia [45,46]. In the first case, the CL band shifts towards the high-energy region [46], and its half-width is quite consistent with our data. The CL spectrum for amorphous thin films is a superposition of two Gaussian components with maxima near 2.0 and 2.6 eV [45], respectively. The authors also report a shift of the emission maximum in the blue range from 2.6 to 2.75 eV, associating it with the different concentration of OH groups in the films synthesized using of two precursor systems in atomic layer deposition. The negligible presence of OH groups in our nanotubes is confirmed by weak IR absorption of 1450–1600 cm^−1^; see discussion in Section 3.2. Indeed, partially hygroscopic samples could adsorb water molecules, and the ionizing radiation from an electron beam could induce the radiolysis of the adsorbed molecules. This effect usually manifests as weak peaks at 3.94, 2.0, and 1.91 eV (315, 620, and 650 nm) in CL spectra, and their intensity is reduced after annealing of the samples [50]. Moreover, HfO_2_ demonstrates a positive Gibbs energy increment, meaning a small moisture-absorption–reaction rate [51]. In our experiments, we detected no contribution of the aforementioned emissions obscured, apparently, by the intensive broad photo- and cathodoluminescence in the samples heated up to 300 and 700 °C.

### 3.5. Photoluminescence Decay

In order to study the kinetic peculiarities of the observed PL, a quantitative analysis of the decay curves was carried out. Emission was recorded at spectral maxima for each sample, such as 2.27 eV (as-grown NTs), 2.4 eV (annealed NTs), and 2.45 eV (nanopowder). Figure 6 shows the measured time dependencies.

The observed PL decay can be described as a superposition of three exponential components:(1)$$I(t)=\sum_{i=1}^{3}A_{i}\cdot e^{-t/\tau_{i}}$$
where *A_i_* are constants; *τ_i_* is decay times, ms. It should be also underscored that the same experimental results can be satisfactorily approximated using Becquerel’s empirical formula:(2)$$I(t)=\frac{I_{0}}{{\left(1+t/C\right)}^{r}}$$
where C is a constant, ms; r is an index whose value is related to the ratio of the effective cross-sections of traps and emission centers [52,53].

The parameters of approximation using the expressions (1) and (2) are presented in Table 3. Similar PL decay times for hafnia nanopowder annealed at 450 °C were obtained in [48]. According to the latter, this temperature provokes a noticeable growth of crystalline domains, and therefore, inclusions of the amorphous phase may explain the similarity of decay times.

The revealed dependencies indicate the non-elementary nature of relaxation and charge transfer processes. In particular, hafnia oxygen vacancies in various configurational and charge states can participate in luminescence mechanisms. The higher the index *r*, estimated for annealed nanotubes, the lower the ratio of the effective cross-sections of traps and emission centers. In other words, the index *r* increases with a decrease in the concentration of defects acting as charge traps at room temperature [52,53]. This gives rise to a faster decay of the PL of annealed NTs in comparison with that of the as-grown nanotubular structures and nanopowder. This is a consequence of annealing in an air atmosphere, due to which the number of oxygen vacancies should diminish.

It is known that there are two types of oxygen vacancies in monoclinic HfO_2_, namely, O_3f_ (with coordination number 3) and O_4f_ (with coordination number 4), which can also exist in different charge states [36,54,55]. According to DFT calculations [36,49], the O_4f_ vacancies are more stable compared to those of O_3f_. In turn, among O_4f_, vacancies V^0^, V^−^, and V^2-^ with two, three, and four electrons, respectively, are the most stable, whereas among O_3f_, fully ionized V^2+^- and V^+^-vacancies are faced [36,55]. These defects originate energy levels in the bandgap as well as near the bottom of the conduction band of hafnia. This fact explains the emergence of a shoulder in the optical absorption spectrum for *hν* > 3 eV. In the PLE spectra, four Gaussian components of 4.6, 4.8, 5.1, and 5.5 eV indicate that photoluminescence is effectively excited via optical transitions to levels located near the bottom of the conduction band; the former may be unoccupied levels of V^−^, V^+^, and V^2+^, correspondingly [55], and the latter may correspond to band-to-band optical transitions.

## 4. Conclusions

In this work, we studied the luminescence features of hafnia nanotubes synthesized via electrochemical oxidation in a potentiostatic regime. It was shown that the nanotubes were grown with a length of 10 ± 3 μm and an average outer diameter of 46 ± 7 nm. According to XRD analysis and Raman and IR spectroscopy, the as-grown NTs had an amorphous structure, further annealing at a temperature of 700 °C, which led to crystallization and their transition to a monoclinic phase.

We measured diffuse reflectance spectra at room temperature, followed by their analysis within the Kubelka-Munk formalism. It was established that a spectral shoulder caused by optically active anion vacancy centers appears in the energy range of *h*ν > 3 eV and distorts the intrinsic absorption edge. Within the modified Tauc approach, we gained estimates of the optical gap width formed by direct allowed band-to-band transitions: *E_g_* = 5.65 ± 0.05 eV and 5.51 ± 0.05 eV for as-grown and annealed nanotubes, respectively. The findings secured correspond to similar characteristics for hafnia in various morphological modifications.

We measured for the first time the cathodoluminescence spectra for HfO_2_ NTs of various phase compositions and also conducted a study of their photoluminescent properties in the temperature range of 10–300 K. It was shown that cooling to 10 K has virtually no influence on the shape of the emission band for the synthesized nanotubes, the maximum of which is in the region of 2.3–2.4 eV. The observed PL and CL are due to processes involving vacancy-based centers in the anionic sublattice of hafnia in various configurational (O_3f_ and O_4f_) and charge states (V^−^, V^+^ and V^2+^).

The PL kinetic curves handle several types of capture centers in nanotubular hafnia. Upon high-temperature annealing up to 700 °C, their concentration diminishes, and a crystalline phase with monoclinic symmetry forms. The revealed regularities may be of practical value in optimizing emission properties for developing luminescent media based on hafnia nanotubes of various phase composition.

## Figures and Tables

**Figure 1 nanomaterials-13-03109-f001:**
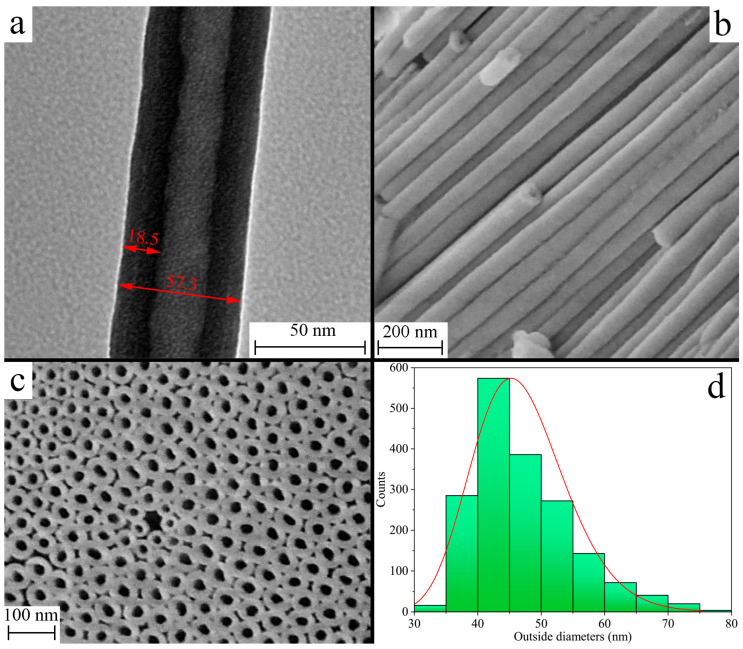
Images of synthesized nanotubes obtained using TEM (**a**), SEM (**b**,**c**). The size distribution of nanotube diameters is shown in (**d**).

**Figure 3 nanomaterials-13-03109-f003:**
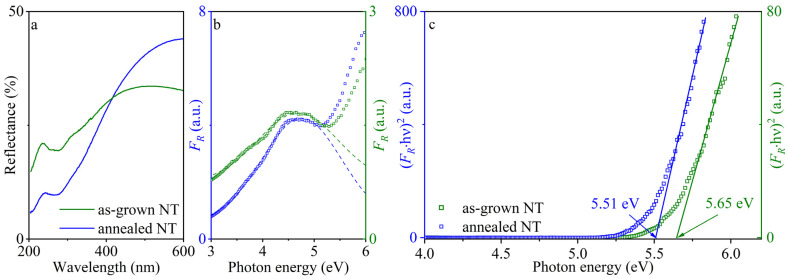
(**a**) The diffuse reflectance spectra measured for as-grown (green) and annealed (blue) nanotubes. (**b**) Absorption spectra *F_R_*(*h*ν) converted using Kubelka–Munk formalism, dashed curves denote a Gaussian-shaped shoulder that emerged because of oxygen vacancy absorption. (**c**) The intrinsic absorption edge plotted in Tauc coordinates for direct allowed transitions taking defect absorption into account.

**Figure 4 nanomaterials-13-03109-f004:**
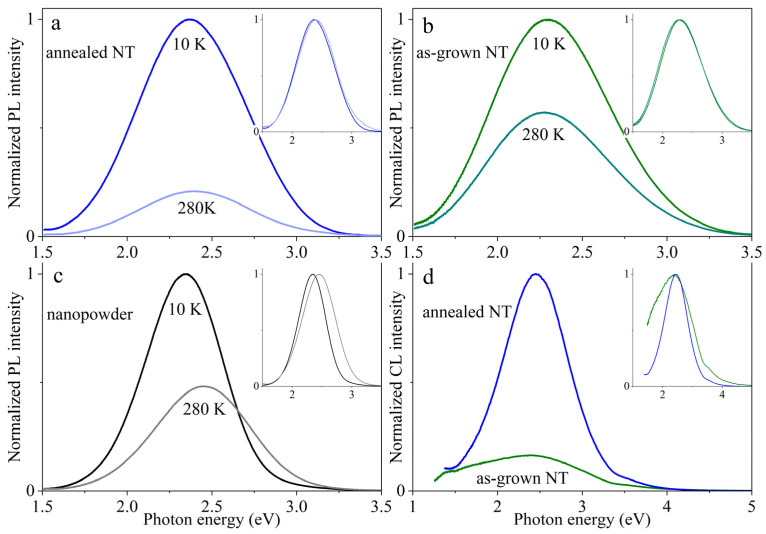
PL spectra for the studied NTs at temperatures of 10 K and 280 K (**a**,**b**), respectively. Spectra for as-grown NTs are shown in green, and spectra for NTs after annealing are shown in blue. For comparison, the PL spectrum of the nanopowder is shown (**c**). Cathodoluminescence spectra (**d**) were measured at room temperature. The spectra normalized to their own maxima for a visual shape comparison are depicted in the insets.

**Figure 5 nanomaterials-13-03109-f005:**
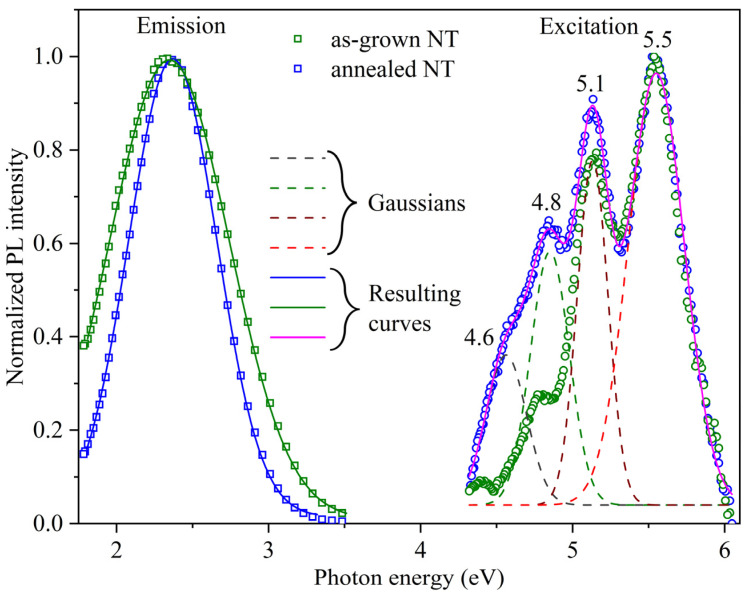
PL emission (squares) and excitation (circles) spectra of as-grown (green) and annealed (blue) NTs. The solid lines indicate the resulting approximations of the measured spectra by individual Gaussian components (dashed lines).

**Figure 6 nanomaterials-13-03109-f006:**
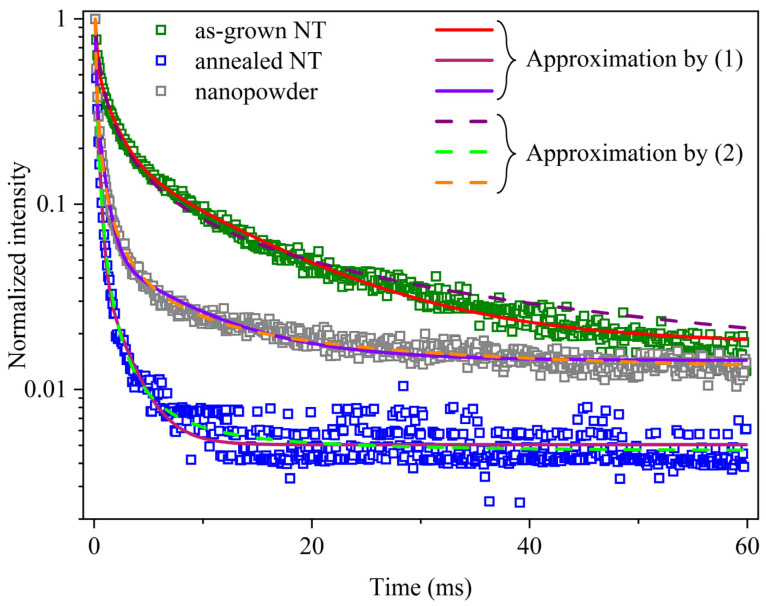
Photoluminescence decay curves in HfO_2_ nanotubes and nanopowder. Experimental data (squares) are approximated using expressions (1) and (2); the approximation curves are represented as solid and dashed lines, respectively.

**Table 1 nanomaterials-13-03109-t001:** Phonon energies of IR-active vibrational modes for the hafnia nanostructures studied.

Peak Energy, cm^−1^			
This Work		Independent Experiment	Calculation [33]
Annealed NTs	Monoclinic Nanopowder		
417	—	410 [28,29] 415 [31]	410, A_u_
428	432	—	?
440	443	—	?
523	523	506 [30] 515 [29] 516 [32]	512, B_u_
543	546	550 [30] 555 [32]	?
602	592	595 [30] 600 [28] 615 [29]	600, A_u_
626	—	625 [29] 635 [28]	?
666	669	680 [32]	665, A_u_
759	759	740 [29] 750 [32] 752 [28]	730, B_u_

“?”—IR maxima which are not assigned to specific vibrational modes in calculations [33].

**Table 2 nanomaterials-13-03109-t002:** Energy parameters for the studied luminescence spectra.

Sample, Excitation Wavelength	Temperature, K	E_max_, ±0.02 eV	FWHM, ±0.02 eV	I_10_/I_280_	Reference
	Photoluminescence				
As-grown nanotubes, 263 nm	10	2.30	0.82	1.75	This work
	280	2.27	0.84		
Annealed nanotubes, 263 nm	10	2.38	0.74	4.78	
	280	2.40	0.75		
Monoclinic nanopowder, 263 nm	10	2.33	0.54	2.07	
	280	2.45	0.66		
Monoclinic nanotubes, 325 nm	RT	2.91 *	0.98 *	—	[13]
Monoclinic nanopowder, 210 nm	RT	2.60	0.58	—	[7]
Monoclinic nanopowder, 200 nm	10	2.96 *	0.91 *	1.05 *	[46]
	RT	2.84 *	0.76 *		
	Cathodoluminescence				
As-grown nanotubes	RT	2.40	1.32	—	This work
Annealed nanotubes		2.45	0.93		
Monoclinic nanopowder		3.00	1.04		[46]
Amorphous films		2.75	0.87		[45]

* Our estimates based on the data given in the cited paper.

**Table 3 nanomaterials-13-03109-t003:** Kinetic parameters of photoluminescence in HfO_2_ nanostructures.

Sample	Multiexponential Decay, See (1)					
	*A* _1_	*A* _2_	*A* _3_	τ_1_, ms	τ_2_, ms	τ_3_, ms
as-grown NTs	0.744	0.382	0.176	0.191	1.56	11.5
annealed NTs	2.595	0.726	0.048	0.056	0.29	2.12
nanopowder	1.777	0.360	0.042	0.096	0.72	7.92
nanopowder [48].	0.284	0.406	0.306	0.198	2.04	11.5
**Sample**	**Becquerel’s decay, see (2)**					
	***C*, ms**			** *r* **		
as-grown NTs	0.357			0.739		
annealed NTs	0.181			1.591		
nanopowder	0.111			0.950		

## Data Availability

The data presented in this study are available on request from the corresponding author.

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
