# Peer review of "Luminescence in Anion-Deficient Hafnia Nanotubes"

_nanomaterials, 2023, doi:10.3390/nano13243109_

Round 1
Reviewer 1 Report
Comments and Suggestions for Authors
See the attached file

Reviewer 2 Report
Comments and Suggestions for Authors
The manuscript is dedicated to the electrochemical synthesis of hafnia nanotubes and their characterization with the emhasis on luminescent properties. The work is performed on a high exerimental level, results are novel and possess significant interest.
The only comment I have is concerning carbon content in as-prepared nanotubes. Authors mentioned that carbon residue is removed by calcination, however, it would be good to estimate the amout of this carbon residue. Is it possible that at least some of the features of defect structure of the final material depends on the said carbon residue and therefore on the preparation conditions (solvent, potential etc.)? This may be important in order to understand, how to achieve the optimal luminescent characteristics and be sure in their reproducibility.
Also minor remark - please uniform the writing of "Ekaterinburg" in affiliations.
